# A Study on the Monitoring of *Toxocara* spp. in Various Children’s Play Facilities in the Republic of Korea (2016–2021)

**DOI:** 10.3390/healthcare11212839

**Published:** 2023-10-27

**Authors:** Young-Hwan Oh, Hae-Jin Sohn, Mi-Yeon Choi, Min-Woo Hyun, Seok-Ho Hong, Ji-Su Lee, Ah-Reum Ryu, Jong-Hyun Kim, Ho-Joon Shin

**Affiliations:** 1Department of Microbiology, Ajou University School of Medicine, Suwon 16499, Republic of Korea; ohyh@tisc.re.kr (Y.-H.O.); hj35good@ajou.ac.kr (H.-J.S.); 2Bio Analysis Team, Testing Institute of Sanitation & Convergence, Cheongju 28463, Republic of Korea; 3Environments & Bio Evaluation Team, FITI Testing & Research Institute, Cheongju 28115, Republic of Korea; mychoi@fiti.re.kr (M.-Y.C.); idhmw84@fiti.re.kr (M.-W.H.); hongsh@fiti.re.kr (S.-H.H.); wltn12533@fiti.re.kr (J.-S.L.); ysjrar1004@fiti.re.kr (A.-R.R.); 4Institute of Animal Medicine, College of Veterinary Medicine, Gyeongsang National University, Jinju 52828, Republic of Korea

**Keywords:** children’s play facilities, *Toxocara* spp., monitoring, the Republic of Korea

## Abstract

*Toxocara* spp. is a zoonotic soil-transmitted parasite that infects canids and felids, which causes toxocariasis in humans, migrating to organ systems, including the lungs, the ocular system, and the central nervous system. Since *Toxocara* spp. is usually transmitted through soil, children tend to be more susceptible to infection. In order to monitor contamination with *Toxocara* spp. in children’s play facilities in the Republic of Korea, we investigated 11,429 samples of soil from daycare centers, kindergartens, elementary schools, and parks across the country from January 2016 to December 2021. Since the Environmental Health Act in the Republic of Korea was enacted in March 2008, there have been sporadic reports of contamination by *Toxocara* spp. in children’s activity zones. In this study, soil from children’s play facilities in regions across the Republic of Korea was monitored according to the Korean standardized procedure to use it as basic data for preventive management and public health promotion. The national average positive rate was 0.16% (18/11,429), and Seoul showed a higher rate of 0.63% (2/318) than any other regions while Incheon, Daegu, Ulsan, Kangwon-do, Jeollabuk-do, and Jeollanam-do were negative (*p* < 0.05). The positive rates were as follows: 0.37% (4/1089) in daycare centers, 0.13% (3/2365) in kindergartens, 0.2% (7/4193) in elementary schools, 0.09% (1/1143) in apartments, and 0.14% (3/2198) in parks. In addition, it was confirmed that 0.2% (1/498) of elementary schools and 1.17% (2/171) of parks were re-contaminated among play facilities managed with the establishment of a regular inspection cycle. Consequently, there is an essential need for continuous monitoring of *Toxocara* spp. contamination and regular education for preschool and school children in order to prevent soil-borne parasite infections.

## 1. Introduction

Soil-transmitted parasites are a wide group of parasites found in the soil [1], and parasite infection is one of the most prevalent clinical conditions in the world. According to the World Health Organization report in 2023, soil-transmitted helminth infections are among the most prevalent infections globally with an estimated 1.5 billion infected people, or 24% of the world’s population [2]. In addition, over 267 million preschoolers and over 568 million schoolchildren live in regions where these parasites are common. Additionally, this is one of the main justifications for the pressing need for parasite treatment and prevention measures [2,3]. Beyond the high rates of mortality and morbidity caused by parasite infections in developing nations, they also pose a serious risk to human health in industrialized nations, affecting people of all ages, especially children [4,5,6]. Among the major species, *Toxocara* spp., in particular, causes the most widespread soil infectious parasite infection worldwide. Soil contamination by *Toxocara* spp. eggs is seen as a severe concern to public health in both developed and developing nations [7]. Ingestion of *Toxocara* spp. eggs in soil is the primary source of human infection, particularly by children, because children are more likely to come into contact with contaminated soil in public spaces like parks, schools, and playground sandboxes [8]. Dogs and cats have apparently played crucial roles in spreading *Toxocara* spp. by shedding eggs, which are the definitive hosts of *Toxocara canis* and *T. cati*, respectively [9]. Human toxocariasis manifests as visceral and ocular larva migrans, eosinophilic meningoencephalitis, covert toxocariasis, and neurological toxocariasis, depending on the quantity of ingested eggs, larval localization, and host reactions [10,11,12,13,14,15,16].

The Environmental Health Act was established by the Korean government in March 2008 in order to create a legal framework in the area of environment and health to address concerns regarding the harms to public health created by various forms of environmental pollution and hazardous chemicals [17]. Environmental safety control standards for children’s activity zones prepared to protect children from environmentally hazardous factors are classified into a total of six categories. The fourth states that soil, such as sand used on the floor of the children’s activity zones, should be managed hygienically according to the standards set by the Ordinance of the Ministry of Environment of the Korean government so that parasites and their eggs are not detected. In addition, inspections should be conducted in accordance with the test procedure of the official test standards for environmentally hazardous factors in the administrative rules of the Environmental Testing and Inspection Act. Other than the official test standards, if a method is recognized domestically and internationally wherein the measurement result is the same or more accurate, it can be applied. A children’s activity zone is a place where children go to play or perform activities, such as daycare centers, kindergartens, primary schools, and kids’ cafes (indoor playgrounds).

In the Republic of Korea, there have been sporadic reports of helminth contamination in kid-friendly activity areas like playgrounds in parks, living areas, and educational facilities. In 1978, a survey was performed concerning soil-transmitted helminth contamination of playgrounds in elementary, middle, and high schools in Masan and Goseong-gun [6]. The highest parasitic contamination rate of 92.5% was found in some educational institutions, though there were some variations between schools. The playgrounds of elementary schools in Incheon were examined for soil-transmitted helminth infection in 1986, and the positive incidence was approximately 7% [18]. The rates of contamination by soil-transmitted helminths in the sand of playgrounds in parks and apartment buildings in the Incheon area were 2.1% and 1.2%, respectively, while in Seoul, 0.4% and 1.15% of these parasites were found in the sand of educational facilities and playgrounds in parks from 2004 to 2008 [19,20]. Soil-transmitted helminth infection still poses a risk despite some variations in the contamination rates recorded by regions and survey periods. The status of soil-transmitted parasite infection has been controlled in some play facilities in each region before and after the legal basis was created. However, few studies focus on play areas across the country. In this study, the soil in playgrounds installed in children’s activity zones, such as daycare centers, kindergartens, elementary schools, and parks, was monitored for *Toxocara* spp. contamination in regions across the Republic of Korea according to the Korean standardized procedure to use it as basic data for preventive management and public health promotion.

## 2. Materials and Methods

### 2.1. Subjects of Investigation

This study was conducted on soil samples across the Republic of Korea over six years from January 2016 to December 2021. A total of 11,429 soil samples from children’s play facilities (1089 daycare centers, 2365 kindergartens, 4193 elementary schools, 1143 apartments, 2198 parks, and 441 local playgrounds) were examined in regions nationwide.

### 2.2. Classification of Samples by Regions

Each region of the Republic of Korea was classified as an administrative district such as Seoul, Sejong, 6 metropolitan cities, 8 provinces, and Jeju Island (Figure 1). A total of 11,429 samples of soil from children’s play facilities were examined according to regions: 318 cases from Seoul, 533 cases from Sejong, 193 cases from Incheon, 3052 cases from Daejeon, 419 cases from Daegu, 571 cases from Ulsan, 285 cases from Busan, 1030 cases from Gyeonggi-do, 250 cases from Gangwon-do, 739 cases from Chungcheongbuk-do, 1088 cases from Chungcheongnam-do, 707 cases from Jeollabuk-do, 224 cases from Jeollanam-do, 1104 cases from Gyeongsangnam-do, 422 cases from Gyeongsangbuk-do, and 401 cases from Jeju-do.

### 2.3. Classification of Samples by Year of Construction

According to the year of construction (including the date of the construction permit or approval), daycare centers, kindergartens, elementary schools, and apartments with children’s play areas were arbitrarily divided into four groups by ten years: Before 1950, 1950 to 2000, 2001 to 2010, and 2011 to 2021. Additionally, if the building year was unclear, it was labeled as Unknown. Facilities that had been asked to be inspected more than once regularly (or occasionally) over a specific period of time, including children’s play areas in daycare facilities, kindergartens, elementary schools, and apartments, were individually investigated. The manager of the play facility acted arbitrarily in cases where the facility frequently sought inspections.

### 2.4. Collection of Samples

Soil samples were collected by obtaining commissions from organizations and schools that manage children’s play facilities in each region or visiting the target location directly. According to Article 6 of the Environmental Testing and Inspection Act in the Korean government, when measuring environmental hazardous factors, we had to follow the official test standards for environmental pollution, which prescribes all matters necessary to ensure the unity and accuracy of measurement. As for the collection method, it was carried out according to the procedure of microscopic examination (ES 12711.1a), which is a method for testing parasites (eggs) in sand and soil according to the official test standards for environmental pollution [21]. The area chosen for the collection location was divided into 5 zones: The center, east, west, south, and north across the country. In each of the five zones, a location with animal droppings in the soil or a location where animals could be seen readily was chosen. Samples were collected in an equal amount of approximately 250 g from the topsoil layer at each collection point to a depth of approximately 15 cm. The collected soil was put in a sample bag; necessary information was completed with regard to factors such as the location of the play facility, collection point, date, and collector; and it was transported to the laboratory. The collected samples were examined as soon as possible, and if not, they were stored at a low temperature of 5 ± 3 °C.

### 2.5. Preparation of Samples

The floating method using zinc sulfate (ZnSO_4_, 161.56) was carried out in accordance with the procedure of microscopic examination, which is a method of testing parasites (eggs) in sand and soil from the official test standards for environmental pollution [21,22,23,24]. First, a 1000 mL beaker was filled with 100 g of the collected material. Thereafter, 500 mL of purified water was added, and the concoction was thoroughly stirred using a glass rod. To eliminate suspended solids, the combined sample was filtered through a testing sieve, and the filtrate was allowed to sit for 30 min. The filtrate was transferred to a separatory funnel and allowed to stand for 30 min after the supernatant was meticulously removed, leaving approximately 200 mL of the filtrate behind. The supernatant was discarded after 50 mL of the sediment liquid from the separatory funnel’s bottom layer was transferred to a centrifuge container and centrifuged at 1500 rpm for two minutes. The leftover lower layer received one more addition of pure water, and the mixture was then centrifuged for two minutes. Centrifugation was carried out at 1500 rpm for one minute and allowed to stand for two to five minutes afterward. The supernatant was discarded, and 50 mL of saturated zinc sulfate solution was added to the lower layer of the liquid. Using a platinum loop, the top layer of the centrifuge tube was removed, put on a slide glass, and then covered with a cover glass to create a slide sample. This slide sample was then examined under an optical microscope (×100–400).

### 2.6. Observation of Samples

With reference to various previous reports [18,19,20,22,23,24], the criterion for egg size was set at 65–90 μm. A sample with at least a single egg detected was considered positive, and the eggs were identified as *Toxocara* spp. eggs based on their morphology. Discrimination between *Toxocara canis* and *Toxocara cati* egg by microscopic examination was based on the ova sizes (85 × 75 μm and 75 × 65 μm, respectively), verified by an ocular micrometer, as well as the shape, color, and wall thickness.

### 2.7. Statistical Analysis

Statistical analyses were performed using the Microsoft Excel program 2016 (Microsoft Office 2016) (Microsoft, Redmond, Washington, DC, USA). The mean positive rate of the entire region and the positive rate of each region were statistically analyzed using a paired *t*-test. A difference of *p* < 0.05 was considered statistically significant.

## 3. Results

### 3.1. Positive Rates of Toxocara spp. by Regions

From 2016 to 2021, a total of 0.16% (18/11,429) of samples were positive for *Toxocara* spp. eggs (Table 1). In terms of the number of cases detected by year, 1 out of 472 cases (0.22%) were positive in 2016, 2 out of 2448 cases (0.09%) in 2017, 1 out of 1670 cases (0.06%) in 2018, 9 out of 2348 cases (0.39%) in 2019, 4 out of 2407 cases (0.17%) in 2020, and 1 out of 2084 cases (0.05%) in 2021. In particular, Seoul showed a higher positive rate of 0.63% (2/318) than any other regions, while Incheon, Daegu, Ulsan, Kangwon-do, Jeollabuk-do, and Jeollanam-do were negative (*p* < 0.05) (Table 1).

### 3.2. Positive Rates of Toxocara spp. by Types of Play Facilities

Each play facility was classified into daycare centers, kindergartens, elementary schools, apartments, parks, and local playgrounds. The total number of positive cases over 6 years were as follows: 4 out of 1089 (0.37%) in daycare centers, 3 out of 2365 (0.13%) in kindergartens, 7 out of 4193 (0.2%) in elementary schools, 1 out of 1143 (0.09%) in apartments, 3 out of 2198 (0.14%) in parks, and 0 out of 442 (0%) in local playgrounds, respectively (*p* < 0.05) (Table 2). The positive rates of each play facility were relatively similar, but they became higher in the order of local playgrounds, kindergartens, parks, elementary schools, and daycare centers (Table 2).

### 3.3. Positive Rates of Toxocara spp. by the Year of Construction

According to the year of construction (including the date of the construction permit or approval), play facilities were divided into four groups: Before 1950, 1950 to 2000, 2001 to 2010, and 2011 to 2021. The contamination rates of *Toxocara* spp. eggs were confirmed in 2 out of 1484 facilities built before 1950 (0.14%), 8 out of 3689 facilities built between 1951 and 2000 (0.22%), 2 out of 1760 facilities built between 2001 and 2010 (0.12%), and 2 out of 1804 facilities built between 2011 and 2021 (0.11%), respectively (*p* < 0.05) (Table 3). The positive rates of play facilities built before 1950 were not very different from those of the play facilities that were built recently. The contamination rates of *Toxocara* spp. Eggs in play facilities did not appear to have any tendency in relation to the year of construction.

### 3.4. Positive Rates of Toxocara spp. in Regular Inspection Areas

In the case of the play facilities that have a regular (or infrequent) inspection cycle set or that have been inspected and managed at least once over a predetermined time period, 3 out of 1105 facilities (0.28%) appeared to be positive for *Toxocara* spp. Eggs (Table 4). Among the play facilities, 1 out of 498 elementary schools (0.2%) and 2 out of 171 parks (1.17%) were confirmed to be re-contaminated, while daycare centers, kindergartens, and apartments were negative (*p* < 0.05) (Table 4).

## 4. Discussion

This study was conducted to measure the level of *Toxocara* spp. Contamination in the soil of children’s play facilities in the Republic of Korea. From January 2016 to December 2021, a total of 11,429 soil samples from children’s play facilities were examined by region across the country, and the study showed 0.16% of the mean positive rate nationwide. Soil-transmitted and zoonotic parasites, such as *Toxocara* spp., remain among the world’s most serious health problems, particularly in developing nations [25]. *Toxocara* spp. Eggs can survive in the environment for years, facilitating transmission [26,27]. When people consume dirt contaminated with animal feces carrying pathogenic *Toxocara* spp. Eggs, they become accidental hosts [28,29]. Furthermore, everyone is susceptible to toxocariasis, and children who play in public places are more likely to contract the infection than adults are [30,31]. Another reason why toxocariasis is frequent in children is because they are likely to play with soil, eat contaminated food with filthy hands, and come into frequent contact with dogs. It is difficult for them to maintain their own personal hygiene [29]. In addition, cats have a habit of burying their feces in the ground, which can lead to parasite contamination in the soil [32].

When the Environmental Health Act was enacted in the Republic of Korea in March 2008, environmental safety management standards for children’s activity zones were established to protect children from harmful environmental factors. Before and after the enactment of the Environmental Health Act, there have been reports that investigated *Toxocara* spp. Eggs in the soil of children’s play facilities in some regions. Since 1986, the positive rates of *Toxocara* spp. in each region of the Republic of Korea have been investigated as follows: 19.2% in playgrounds of elementary school in Incheon [18], 18.5% in children’s parks of detached houses in Daejeon in 2000 [33], 9.2% in playgrounds of apartment complexes and 30.1% in public parks in Incheon from 2004 to 2008 [19], and 0.88% in children’s parks and educational facilities in Seoul in 2013 [20]. In our study, to assess the real level of *Toxocara* spp. Egg contamination of the soil in play facilities of children’s activity zones of the Republic of Korea, such as daycare centers, kindergartens, elementary schools, parks, and local playgrounds, a total of 11,429 cases were investigated according to region from 2016 to 2021. As a result, 0.16% of 11,429 cases in play facilities nationwide were positive for *Toxocara* spp. Eggs.

The cases reported in other countries included the following. Within the East-Midlands region in England, the risks of being infected posed to those regularly frequenting the public parks, where the soil was contaminated with *Toxocara* spp. [34]. In Brazil, it was observed that the number of dogs frequently visiting parks affected soil pollution in public spaces [35]. Additionally, it was reported that *Toxocara* spp. eggs were most easily detected in backyard soil samples in Poland [36] and from sandpits in Japan [37]. *Toxocara* spp. eggs were substantially more common in urban regions than in suburban or rural locations [38,39,40]. According to some research, *Toxocara* spp.-contaminated parks were not evenly distributed throughout New York City but were more common in neighborhoods with lower socioeconomic levels [41]. In this study, urban areas such as Seoul showed slightly higher positive rates than other areas, but no clear difference between urban and rural areas was confirmed. Many factors, such as sampling locations, sampling depth, number and volume of samples, recovery method, inspection season, type of soil surveyed, preservation of samples (time and conditions), and laboratory capabilities, can affect soil test results during sampling and laboratory analysis in the case of regional differences in pollution levels of each play facility by region [42,43,44]. Moreover, soil survey methods are not standardized, making it difficult to compare specific survey results [45]. Therefore, it can be challenging to compare the findings of different reports and determine what implications they have for public health due to the lack of technological standardization and the abundance of factors influencing the process of sampling eggs. These factors may also lead to false-negative results and an underestimation of the prevalence of contamination [44].

Meanwhile, *Toxocara* spp. eggs have a thick shell that gives them the ability to endure the severe conditions of the soil, chemical wastewater treatment, and sewage sludge for long periods of time [10,15,46,47]. In this study, it was expected that the older the building year, the higher the survival rate of *Toxocara* spp. eggs, but no significant tendency was confirmed. However, since the scope of this survey was widespread, it is considered necessary to conduct further research taking the differences in region, environment, etc., into account.

In the case of housing types in the Republic of Korea, apartment housing types are rising compared to detached houses. Children living in apartment complexes naturally spend more time at play facilities in their apartment complexes than in other places. Therefore, play facilities of playgrounds, kindergartens, and daycare centers operated in apartment complexes require more special management because they easily become the specific places where children can come into frequent contact with the soil. Soil contamination with *Toxocara* spp. eggs has been reported worldwide, but no active countermeasures to prevent *Toxocara* spp. transmission from pets to humans have been proposed. Washing hands after playing in the sand, removing internal parasites from pets, enclosing sandpits, educating pet owners, and limiting dog roaming were all suggested as preventative methods in early studies of toxocariasis [48,49,50,51]. However, there is no evidence of the effectiveness of these actions. Schantz [48] also suggested that children not be allowed to play in places polluted with *Toxocara* spp. eggs, which was impossible because most playgrounds were already contaminated.

In the case of the Republic of Korea, it is stipulated that the operators of children’s play facilities must manage the sands hygienically since the legal basis has been established. As a result of interviewing the people in charge of the daycare centers, kindergartens, and elementary schools where their facilities are managed by the regular inspection cycle, they said that the play facilities with positive results have been managed mainly by disinfection and consignment and improved by steam disinfection, general sterilization, and replacement of sand. Although the frequency is different for each local government, they have been carrying out sand disinfection once to four times a year in order to provide pleasant activity zones for children. Among the various disinfection methods, the main method uses a steam sterilizer, which boils water above 150 °C and then sprays high-pressure steam at a depth of more than 10 cm in the soil. Nevertheless, there are far more open play facilities than those with appropriate preventive devices, such as installing surrounding fences and protection covers for sands. These open play facilities are often re-contaminated due to free access to roaming dogs or cats and wild rodents. Finally, as a result of this study, the contamination rates of *Toxocara* spp. eggs were observed in play facilities where preschoolers and schoolchildren mainly hang out, so it is necessary to conduct continuous monitoring and regular personal hygiene education to prevent infection.

## 5. Conclusions

As a result of this study, it was verified that the infection of *Toxocara* spp. eggs still poses a risk in regional children’s play facilities across the Republic of Korea. Therefore, there is an essential need for appropriate control through continuous monitoring of *Toxocara* spp. and regular personal hygiene education for children to prevent infections in children’s play facilities.

## Figures and Tables

**Figure 1 healthcare-11-02839-f001:**
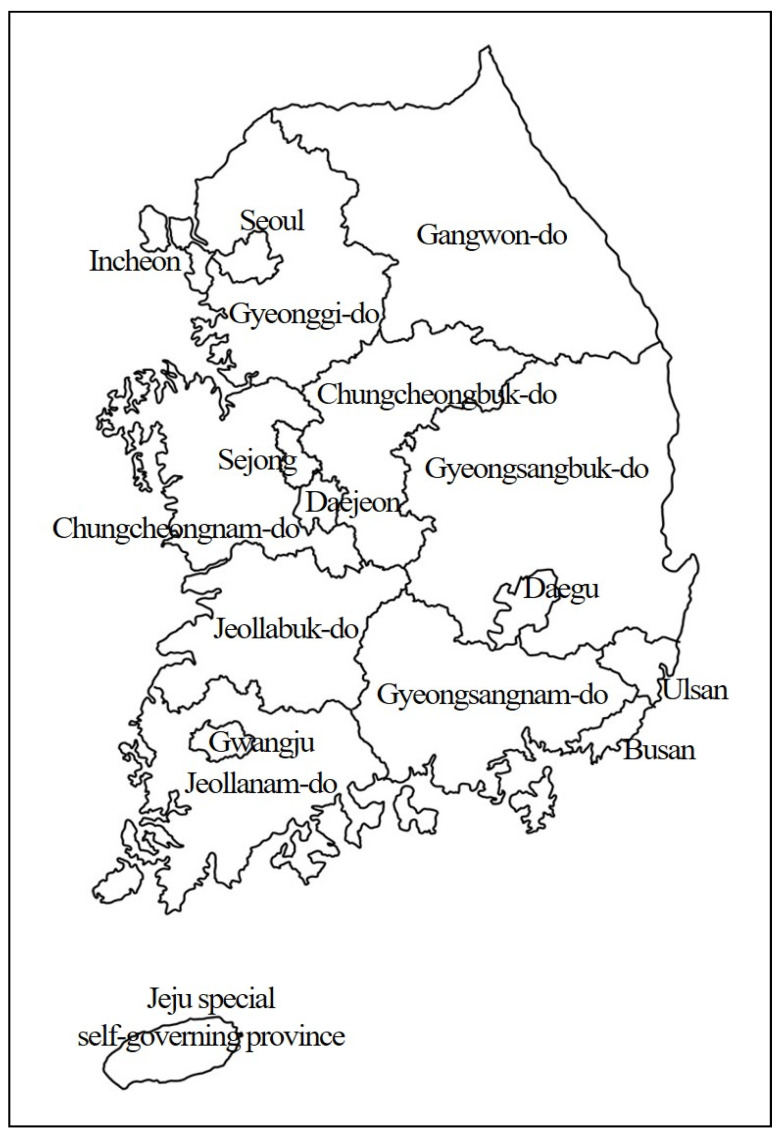
The map of the administrative divisions of the Republic of Korea shows one Seoul metropolitan government, one Sejong special self-governing city, six metropolitan cities, eight provincial areas, and one Jeju special self-governing province where the sampled play facilities were located.

**Table 1 healthcare-11-02839-t001:** Monitoring detection of *Toxocara* spp. eggs collected from play facilities in each region nationwide (2016–2021).

Survey Area	No. Positive/No. Examined (%)						
	2016	2017	2018	2019	2020	2021	Total
Seoul	0/19 (0)	0/34 (0)	0/87 (0)	0/39 (0)	2/41 (4.88)	0/98 (0)	2/318 (0.63) *
Sejong	0/20 (0)	0/92 (0)	0/63 (0)	1/218 (0.46)	0/58 (0)	0/82 (0)	1/533 (0.19) *
Incheon	0/5 (0)	0/11 (0)	0/9 (0)	0/17 (0)	0/135 (0)	0/16 (0)	0/193 (0)
Daejeon	0/190 (0)	0/429 (0)	0/503 (0)	2/610 (0.33)	0/646 (0)	0/675 (0)	2/3052 (0.07) *
Daegu	0/2 (0)	0/233 (0)	0/13 (0)	0/26 (0)	0/61 (0)	0/84 (0)	0/419 (0)
Ulsan	0/0 (0)	0/30 (0)	0/18 (0)	0/16 (0)	0/16 (0)	0/13 (0)	0/93 (0)
Gwangju	0/3 (0)	0/285 (0)	1/86 (1.17)	1/111 (0.90)	0/78 (0)	0/8 (0)	2/571 (0.35) *
Busan	0/24 (0)	0/96 (0)	0/31 (0)	1/47 (2.13)	0/57 (0)	0/30 (0)	1/285 (0.35) *
Gyeonggi-do	0/58 (0)	0/206 (0)	0/167 (0)	0/185 (0)	1/206 (0.49)	0/208 (0)	1/1030 (0.10) *
Gangwon-do	0/4 (0)	0/10 (0)	0/73 (0)	0/63 (0)	0/34 (0)	0/66 (0)	0/250 (0)
Chungcheongbuk-do	0/22 (0)	0/92 (0)	0/90 (0)	1/187 (0.54)	0/219 (0)	0/129 (0)	1/739 (0.14) *
Chungcheongnam-do	0/10 (0)	0/282 (0)	0/311 (0)	0/146 (0)	0/154 (0)	0/185 (0)	0/1088 (0)
Jeollabuk-do	0/30 (0)	0/249 (0)	0/54 (0)	0/129 (0)	0/122 (0)	0/123 (0)	0/707 (0)
Jeollanam-do	0/14 (0)	0/30 (0)	0/20 (0)	0/37 (0)	0/66 (0)	0/57 (0)	0/224 (0)
Gyeongsang nam-do	1/65 (1.54)	2/349 (0.58)	0/104 (0)	2/188 (1.07)	0/255 (0)	0/143 (0)	5/1104 (0.46) *
Gyeongsang buk-do	0/7 (0)	0/19 (0)	0/41 (0)	0/116 (0)	1/150 (0.67)	0/89 (0)	1/422 (0.24) *
Jeju Island	0/0 (0)	0/1 (0)	0/0 (0)	1/213 (0.47)	0/109 (0)	1/78 (1.29)	2/401 (0.50) *
Total	1/472 (0.22)	2/2448 (0.09)	1/1670 (0.06)	9/2348 (0.39)	4/2407 (0.17)	1/2084 (0.05)	18/11,429 (0.16)

* Asterisks indicate statistical significance (*p* < 0.05).

**Table 2 healthcare-11-02839-t002:** Monitoring status of *Toxocara* spp. eggs detection in each play facility by year.

Collection Places	No. Positive/No. Examined (%)						
	2016	2017	2018	2019	2020	2021	Total
Daycare centers	1/48 (2.09)	0/344 (0)	0/156 (0)	2/141 (1.42)	0/211 (0)	1/189 (0.53)	4/1089 (0.37) *
Kindergartens	0/70 (0)	0/530 (0)	1/383 (0.27)	2/455 (0.44)	0/458 (0)	0/469 (0)	3/2365 (0.13) *
Elementary Schools	0/192 (0)	2/845 (0.24)	0/549 (0)	3/923 (0.33)	2/944 (0.22)	0/740 (0)	7/4193 (0.2) *
Apartments	0/68 (0)	0/446 (0)	0/194 (0)	0/126 (0)	1/139 (0.72)	0/170 (0)	1/1143 (0.09) *
Parks	0/66 (0)	0/224 (0)	0/346 (0)	2/618 (0.33)	1/529 (0.19)	0/415 (0)	3/2198 (0.14) *
Local playgrounds	0/28 (0)	0/59 (0)	0/42 (0)	0/85 (0)	0/126 (0)	0/101 (0)	0/441 (0)

* Asterisks indicate statistical significance (*p* < 0.05).

**Table 3 healthcare-11-02839-t003:** Monitoring status of *Toxocara* spp. Eggs detection in each play facility by the year of construction.

Play Facilities	No. Positive/No. Examined (%)				
	Before 1950	1951–2000	2001–2010	2011–2021	Unknown
Daycare centers	0/5 (0)	2/283 (0.71)	0/318 (0)	1/463 (0.22)	0/20 (0)
Kindergartens	0/58 (0)	1/1155 (0.09)	2/516 (0.39)	0/612 (0)	0/24 (0)
Elementary schools	2/1421 (0.14)	4/1569 (0.26)	0/645 (0)	1/549 (0.19)	0/5 (0)
Apartments	0/0 (0)	1/682 (0.15)	0/281 (0)	0/180 (0)	0/0 (0)
Total	2/1484 (0.14) *	8/3689 (0.22) *	2/1760 (0.12) *	2/1804 (0.11) *	0/49 (0)

* Asterisks indicate statistical significance (*p* < 0.05).

**Table 4 healthcare-11-02839-t004:** Monitoring status of *Toxocara* spp. Eggs detection in each play facility managed with the establishment of regular inspection cycles.

No. Positive/No. Examined (%)					
Daycare Centers	Kindergartens	Elementary Schools	Apartments	Parks	Total
0/111 (0)	0/276 (0)	1/498 (0.2) *	0/49 (0)	2/171 (1.17) *	3/1105 (0.28)

* Asterisks indicate statistical significance (*p* < 0.05).

## Data Availability

The data are available upon request from the corresponding author.

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
