# Peer review of "A Study on the Monitoring of Toxocara spp. in Various Children’s Play Facilities in the Republic of Korea (2016–2021)"

_healthcare, 2023, doi:10.3390/healthcare11212839_

Round 1

Reviewer 1 Report

Comments and Suggestions for Authors

The manuscript addresses a very important topic of Toxocara infection in children due to contaminated soil in play facilities. The experimental design and the results presented are adequate, however the following points may be considered:

1. English language and grammar need minor corrections

2. The possibility of infection in children due to cross-contamination could also have been considered

3. It is not clear if the samples were collected from childrens' hands and mouth

Comments on the Quality of English Language

English language and grammar of the manuscript require minor corrections.

Author Response

  1. Reviewer 1

Comments and Suggestions for Authors

The manuscript addresses a very important topic of Toxocara infection in children due to contaminated soil in play facilities. The experimental design and the results presented are adequate, however the following points may be considered:

Q1) English language and grammar need minor corrections

A1) We agreed with your comment. According to your comment, we have uploaded a revised version edited through ‘Language Editing Services’ from MDPI proposed by English Editing Department. (We attached file of the English-Editing-Certificate.)

Q2) The possibility of infection in children due to cross-contamination could also have been considered

A2) Thank you for your suggestion. As your comment, children can be exposed to and cross-infected with various parasites or bacteria from outdoor play facilities. This study was conducted to confirm the presence of Toxocara spp. infection in outdoor play facilities mainly used by children. Based on these results, it will be necessary to investigate whether the children are actually infected with Toxocara spp. in the further study.

Q3) It is not clear if the samples were collected from childrens' hands and mouth

A3) Thank you for your comment. This study was conducted to confirm the presence of Toxocara spp. infection in outdoor play facilities mainly used by children. Accordingly, all the samples of this study were collected from the soil in children’s play facilities.

Comments on the Quality of English Language

Q4) English language and grammar of the manuscript require minor corrections.

A4) We agreed with your comment. According to your comment, we have uploaded a revised version edited through ‘Language Editing Services’ from MDPI proposed by English Editing Department. (We attached file of the English-Editing-Certificate.)

Reviewer 2 Report

Comments and Suggestions for Authors

The study titled "A Study on the Monitoring of Toxocara spp. in Various Children’s Play Facilities in the Republic of Korea (2016-2021)" raises public awareness about the importance of maintaining clean and safe play environments for the children of the Republic of Korea. It highlights the need for regular monitoring and maintenance of play facilities to reduce the risk of parasitic infections and other health related concerns. This study is significant as it addresses a critical issue related to child health and safety, provides valuable data for policy development and public awareness, and contributes to the broader knowledge of parasitology and epidemiology. It has the potential to improve the overall well-being of children and communities by promoting safe and healthy play environments.

Concerning the article's quality, there exist a multitude of typographical and grammatical inaccuracies, along with technical writing problems within the manuscript. I would recommend that the author thoroughly review the manuscript to address these issues and consider possible revisions.

Some of the minor/major concerns are as follows:

Overall English quality of the manuscript is very poor.

Abstract: Add statistical values in the abstract, where applicable

Keywords: children's play facilities; Toxocara spp.; monitoring, Republic of Korea.

Introduction: Add information on Pathogenesis of Toxocara spp.

Line No. 32-34: Rrephrase the sentence.

Materials and Methods:

What criteria was used to collect the specific number of samples from the study regions?

The Materials and Methods section of the manuscript lacks any details or information pertaining to the statistical analysis conducted in the study. Moreover, upon reviewing the Abstract, Results, and Discussion sections, it appears that the results were not presented based on statistical analysis. Merely discussing the results based on percentage values, without taking into account the statistical analysis, lacks scientific rigor and will not provide a meaningful contribution to the scientific community.

References: The majority of the references cited are dated, primarily from earlier years. Please consider incorporating more recent citations from the period spanning 2018 to 2023.

Comments on the Quality of English Language

Overall English quality of the manuscript is very poor.

Author Response

2. Reviewer 2

Comments and Suggestions for Authors

The study titled "A Study on the Monitoring of Toxocara spp. in Various Children’s Play Facilities in the Republic of Korea (2016-2021)" raises public awareness about the importance of maintaining clean and safe play environments for the children of the Republic of Korea. It highlights the need for regular monitoring and maintenance of play facilities to reduce the risk of parasitic infections and other health related concerns. This study is significant as it addresses a critical issue related to child health and safety, provides valuable data for policy development and public awareness, and contributes to the broader knowledge of parasitology and epidemiology. It has the potential to improve the overall well-being of children and communities by promoting safe and healthy play environments.

Concerning the article's quality, there exist a multitude of typographical and grammatical inaccuracies, along with technical writing problems within the manuscript. I would recommend that the author thoroughly review the manuscript to address these issues and consider possible revisions.

Some of the minor/major concerns are as follows:

Q1) Overall English quality of the manuscript is very poor.

A1) According to your comment, we have uploaded a revised version edited through ‘Language Editing Services’ from MDPI proposed by English Editing Department. (We attached file of the English-Editing-Certificate.)

Q2) Abstract: Add statistical values in the abstract, where applicable

A2) Thank you for your comment. We completely agree with you. According to your comment, statistical analysis and values used in this study has been added to ‘Abstract’, ‘Materials and Methods’, and ‘Results’ respectively in manuscript. (line 24, 168, 182, 191, 201, 213)

Q3) Keywords: children's play facilities; Toxocara spp.; monitoring, Republic of Korea.

A3) Thank you for your suggestion. According to your comment, we changed the keywords.

Q4) Introduction: Add information on Pathogenesis of Toxocara spp.

A4) Thank you for your comment. According to your comment, we have added information on pathogenesis of Toxocara spp. in ‘Introduction’ of manuscript. (line 48-53, ref 8, 9)

Q5) Line No. 32-34: Rrephrase the sentence.

A5) Thank you for your comment. We revised the sentence. (line 36-39)

Materials and Methods:

Q6) What criteria was used to collect the specific number of samples from the study regions?

A6) Due to the broad scope of this study, samples were collected in two ways. One was collecting samples by obtaining commissions from organizations that manage children’s play facilities, and the other was by visiting the target location directly. The area chosen for the collection location was divided into 5 zones: the center, east, west, south, and north across each regions. The number of children's play facilities was different in each region, and the number of samples from the study regions was bound to vary, depending on whether the managers of each children's play facility agreed to sample collection.

Q7) The Materials and Methods section of the manuscript lacks any details or information pertaining to the statistical analysis conducted in the study. Moreover, upon reviewing the Abstract, Results, and Discussion sections, it appears that the results were not presented based on statistical analysis. Merely discussing the results based on percentage values, without taking into account the statistical analysis, lacks scientific rigor and will not provide a meaningful contribution to the scientific community.

A7) Thank you for your comment. We completely agree with you. According to your comment, statistical analysis and values used in this study has been added to ‘Abstract’, ‘Materials and Methods’, and ‘Results’ respectively in manuscript. (line 24, 168, 182, 191, 201, 213)

Q8) References: The majority of the references cited are dated, primarily from earlier years. Please consider incorporating more recent citations from the period spanning 2018 to 2023.

A8) Thank you for your suggestion. According to your comment, we added some references (reported in 2018 to 2023) in revised manuscript. (ref 8, 9, 15, 34, 41, 46, 47)

Comments on the Quality of English Language

Q9) Overall English quality of the manuscript is very poor.

A9) According to your comment, we have uploaded a revised version edited through ‘Language Editing Services’ from MDPI proposed by English Editing Department. (We attached file of the English-Editing-Certificate.)

Reviewer 3 Report

Comments and Suggestions for Authors

There are some minor issues. Discussion section should be started with a brief report of the results and findings of the authors. This applies to the Abstract section: authors need to add 1-2 lines background and what Toxocara spp. is and why this infection is important. Not all readers are experts of this type of infection.  

Author Response

  1. Reviewer 3

Comments and Suggestions for Authors

Q1) There are some minor issues. Discussion section should be started with a brief report of the results and findings of the authors. This applies to the Abstract section: authors need to add 1-2 lines background and what Toxocara spp. is and why this infection is important. Not all readers are experts of this type of infection. 

A1) Thank you for your comment. We completely agree with your comment. According to your comment, we have revised some parts of ‘Abstract’ and ‘Discussion’ sections. (line 12-15, 217-220)

Round 2

Reviewer 2 Report

Comments and Suggestions for Authors

I commend the authors for their efforts in enhancing the quality of the manuscript. The substantial improvements made have strengthened the scientific validity of the article. 

However, I suggest the authors to include statistical values within the tabular data.

Comments on the Quality of English Language

English language quality is fine

Author Response

[Responses to reviewer’s comments]

Reviewer 2

Comments and Suggestions for Authors

I commend the authors for their efforts in enhancing the quality of the manuscript. The substantial improvements made have strengthened the scientific validity of the article.

However, I suggest the authors to include statistical values within the tabular data.

A) Thank you for your suggestion. According to your suggestion, we have included statistical values within tables and added a sentence to the footnote of tables in revised manuscript. (Table 1–4)

* All revised words and sentences were highlighted in yellow in the manuscript.
